# Development and Validation of Bioanalytical LC–MS/MS Method for Pharmacokinetic Assessment of Amoxicillin and Clavulanate in Human Plasma

**DOI:** 10.3390/ph18070998

**Published:** 2025-07-02

**Authors:** Sangyoung Lee, Da Hyun Kim, Sabin Shin, Jee Sun Min, Duk Yeon Kim, Seong Jun Jo, Ui Min Jerng, Soo Kyung Bae

**Affiliations:** 1College of Pharmacy and Integrated Research Institute of Pharmaceutical Sciences, The Catholic University of Korea, Bucheon 14662, Republic of Korea; nssy0416@catholic.ac.kr (S.L.); gmkdh2@catholic.ac.kr (D.H.K.); tkqls3840@catholic.ac.kr (S.S.); sunny08@catholic.ac.kr (J.S.M.); keny960508@catholic.ac.kr (D.Y.K.); seongjun@buffalo.edu (S.J.J.); 2Department of Pharmaceutical Sciences, State University of New York, Buffalo, NY 14215, USA; 3Chungwoo Korean Medicine Hospital, Hanam 12919, Republic of Korea; healmind@paran.com

**Keywords:** amoxicillin, clavulanate, human K_2_-EDTA plasma, LC–MS/MS, validation, pharmacokinetics

## Abstract

**Background/Objectives**: We developed and validated a robust and simple LC–MS/MS method for the simultaneous quantification of amoxicillin and clavulanate in human plasma relative to previously reported methods. **Methods**: Amoxicillin; clavulanate; and an internal standard, 4-hydroxytolbutamide, in human K_2_-EDTA plasma, were deproteinized with acetonitrile and then subjected to back-extraction using distilled water–dichloromethane. Separation was performed on a Poroshell 120 EC-C_18_ column with a mobile-phase gradient comprising 0.1% aqueous formic acid and acetonitrile at a flow rate of 0.5 mL/min within 6.5 min. The negative electrospray ionization modes were utilized to monitor the transitions of *m*/*z* 363.9→223.1 (amoxicillin), *m*/*z* 198.0→135.8 (clavulanate), and *m*/*z* 285.0→185.8 (4-hydroxytolbutamide). **Results/Conclusions**: Calibration curves exhibited linear ranges of 10–15,000 ng/mL for amoxicillin (*r* ≥ 0.9945) and 20–10,000 ng/mL for clavulanate (*r* ≥ 0.9959). Intra- and inter-day’s coefficients of variation, indicating the precision of the assay, were ≤7.08% for amoxicillin and ≤10.7% for clavulanate, and relative errors in accuracy ranged from −1.26% to 10.9% for amoxicillin and from −4.41% to 8.73% for clavulanate. All other validation results met regulatory criteria. Partial validation in lithium–heparin, sodium–heparin, and K_3_-EDTA plasma confirmed applicability in multicenter or large-scale studies. This assay demonstrated itself to be environmentally friendly, as assessed by the Analytical GREEnness (AGREE) tool, and was successfully applied to a clinical pharmacokinetic study of an Augmentin^®^ IR tablet (250/125 mg). The inter-individual variabilities in clavulanate exposures (AUC_t_ and C_max_) were significantly greater than in amoxicillin, and they may inform the clinical design of future drug–drug interaction.

## 1. Introduction

Amoxicillin (7-[2-amino-2-(4-hydroxyphenyl)-acetyl]amino-3,3-dimethyl-6-oxo-2-thia-5-azabicyclo[3.2.0]heptane-4-carboxylic acid), a β-lactam antibiotic of the penicillin class, is commonly utilized in the treatment of various bacterial infections, including those of the lower and upper respiratory tracts, as well as conditions such as tonsillitis, bronchitis, sinusitis, and pneumonia [1]. Clavulanate ((2R,5R,Z)-3-(2-Hydroxyethylidene)-7-oxo-4-oxa-1-aza-bicyclo[3.2.0]heptane-2-carboxylic acid), when combined with amoxicillin, serves to enhance the efficacy of treatment against bacterial infections by inhibiting β-lactamase enzymes produced by resistant bacteria. This combination protects against the hydrolysis of amoxicillin by β-lactamases [2] and exhibits a synergistic antibacterial effect.

Amoxicillin/potassium clavulanate, sold under the brand name Augmentin^®^ is a widely used antibacterial combination product. Augmentin^®^ is currently available in various formulations, including immediate- and extended-release tablets, chewable tablets, oral suspension, and injectable intravenous solutions in over 150 countries worldwide. In addition, several different presentations of Augmentin^®^ with increasing ratios of amoxicillin–clavulanate are approved for oral (2:1, 4:1, 7:1, 8:1, 14:1, and 16:1 [3,4,5]) and intravenous (5:1 and 10:1 [6,7]) use by adults and pediatric patients. Immediate-release (IR) tablets are available in three strengths: 250/125 mg and 500/125 mg, typically administered every 8 and 12 h; and 875/125 mg, administered every 12 h [8].

Therefore, the simultaneous monitoring of amoxicillin and clavulanate in biospecimens after oral administration of Augmentin^®^ in patients is an important factor when evaluating its safety and efficacy and considering its potential clinical use. To the best of our knowledge, a number of methods leveraging different analytical techniques, including HPLC [9,10], LC–MS [11,12], and LC–MS/MS [13,14,15,16,17,18,19,20,21,22], have been published to simultaneously quantify amoxicillin and clavulanate in biological fluids. These techniques, however, employ relatively large sample volumes (0.45−0.5 mL) [10,13]; relatively long run times (~10 min) [9,10,18]; have a lack of sensitivity [9,11,14,18,21]; and include time-consuming and/or laborious solid-phase extraction [15,16,17], and deproteinization followed by liquid–liquid extraction with evaporation procedure [19], thus constraining the throughput capacity in bioanalytical workflows. In addition, some methods lack full validation according to regulatory guidelines and cannot be directly applied to the bioanalysis [9,10,11,12,13,14,15,16,18,20,21,22]. The brief summary of the simultaneous quantification methods for amoxicillin and clavulanate in biological samples is presented in Table 1. Furthermore, it was reported that β-lactam antibiotics, especially clavulanate, in aqueous solutions are known to be affected by pH, temperature, and ionic strength [23,24]. Due to these instabilities of clavulanate, there is still a need to establish a more convenient method for simultaneous determination of amoxicillin and clavulanate.

In this study, we developed and validated a simple and sensitive LC–MS/MS method that simultaneously quantifies amoxicillin and clavulanate in human K_2_-EDTA plasma, comprising deproteinization with acetonitrile, and then subjected it to back-extraction using distilled water and dichloromethane. We also partially validated this assay using human sodium heparinized, lithium heparinized, and K_3_-EDTA plasma to provide useful information in multicenter or large-scale clinical studies. Finally, this analytical method was efficiently able to characterize pharmacokinetics for amoxicillin and clavulanate in healthy Korean subjects after a single oral administration of Augmentin^®^ IR tablet (250/125 mg) (Ilsung IS Co., Ltd., Seoul, Republic of Korea).

Recent years have seen a growing emphasis on the principles of Green Analytical Chemistry (GAC), which aims to minimize the environmental impact of analytical methods [25,26,27]. To align with the principles of GAC, efforts were made to design with a focus on waste minimization, limited reagent toxicity, and energy-efficient apparatus.

## 2. Results and Discussion

### 2.1. LC–MS/MS Optimization

The mass parameters of the electrospray ionization conditions for amoxicillin, clavulanate, and the internal standard (IS) were tuned in both positive/negative-ion modes. Amoxicillin could be well ionized in both positive and negative modes; in contrast with amoxicillin, higher responses for clavulanate and IS were achieved only in the negative mode. Therefore, precursor ions of [M − H]^−^ at *m*/*z* 363.9 for amoxicillin, *m*/*z* 198.0 for clavulanate, and *m*/*z* 285.0 for the IS were selected in MS/MS mode to identify respective product ions (Figure 1). The most intense peak observed in the MS/MS scan was used to quantify amoxicillin, clavulanate, and the IS. Finally, mass parameters were optimized to achieve the stable and most abundant signal of their specific product ions. Chemical structures and the product ion mass spectra of amoxicillin, clavulanate, and the IS are depicted in Figure 1.

Chromatographic conditions, such as several columns, like commercially available C_18_ and C_8_; different mobile phase conditions, and different flow rates, were screened to achieve higher peak response, better chromatographic separations, and appropriate peak times. The Poroshell 120 EC-C_18_ column (100 × 4.6 mm, 2.7 μm) with a gradient mobile phase consisting of 0.1% aqueous formic acid–acetonitrile at a flow rate of 0.50 mL/min resulted in better peak shapes with high responses for amoxicillin, clavulanate, and 4-hydroxytolbutamide. Under these conditions, all analytes elute in <5 min (run time, 6.5 min/sample), allowing for high-throughput bioanalyses.

To simultaneously detect amoxicillin and clavulanate in plasma, various protein precipitation reagents—including trifluoroacetic acid, trichloroacetic acid, perchloric acid, acidified acetonitrile, acetonitrile, methanol, and their mixtures—were evaluated. Due to the low peak abundance of clavulanate following protein precipitation with acidified reagents and/or methanol, these were excluded from candidates. As a result of these findings, pure acetonitrile was selected as the protein precipitation reagent. Subsequently, a back-extraction step using distilled water was introduced to enhance sensitivity through analytes’ concentration. Ultimately, amoxicillin, clavulanate, and the IS were protein-precipitated from 100 μL of human plasma using 300 μL of ice-cold acetonitrile, and back-extracted using distilled water–dichloromethane (100 μL:350 μL, *v*/*v*). This protein precipitation, followed by back-extraction, simplified the sample preparation procedure and provided excellent reproducibility and sensitivity without the need for a drying step. Notably, the ratio of distilled water to dichloromethane served as a determining factor for the increased sensitivity and best peak shapes of clavulanate. In particular, the water phase was crucial for obtaining good sensitivity for clavulanate. A plasma–acetonitrile volume ratio of 1:3 with the addition of distilled water–dichloromethane (100 μL:350 μL, *v*/*v*) yielded more consistent and higher sensitivity for amoxicillin and clavulanate, especially clavulanate, than the other precipitants. Acetonitrile, distilled water, and dichloromethane were selected as ternary components for deproteinization coupled with back-extraction in this experiment.

In LC–MS/MS analysis, selecting an IS with proper retention time, satisfactory extraction recovery, and controlled matrix effect is highly desirable. For an assay intended to analyze human plasma from multicenter or large-scale clinical studies, the selection of IS should exclude commonly prescribed and self-medication drugs [28]. These criteria suggest that the ideal IS should be either a structural analog or a stable isotope-labeled version of the analyte. However, due to the instability of clavulanate in working solutions left at room temperature for over 6 h, we did not use a stable isotope or chemically similar agent as the IS. Several potential internal standards were tested, and 4-hydroxytolbutamide was chosen as the most suitable.

### 2.2. Method Validation

#### 2.2.1. Specificity and Sensitivity

There were no significant endogenous interferences at the respective retention times of amoxicillin (3.26 min), clavulanate (3.01 min), and the IS (4.29 min) following testing of eight different batches of human plasma. Figure 2 presents typical LC–MS/MS chromatograms of amoxicillin (i), clavulanate (ii), and the IS (iii): blank human plasma (Figure 2a), blank plasma spiked with both analytes at the lower limit of quantification (LLOQ) levels, and IS (Figure 2b), and 2 h plasma sample after a single oral administration of Augmentin^®^ (250/125 mg) (Figure 2c).

The LLOQs for amoxicillin and clavulanate were determined to be 10 and 20 ng/mL, respectively, with a signal-to-noise ratio > 10, satisfying the criteria for clinical pharmacokinetics.

#### 2.2.2. Linearity

A 1/*x*^2^ weighted regression model was applied to best fit the concentration–response relationship and assessed the linearity of the calibration curves for each analyte. The calibration curves for amoxicillin and clavulanate demonstrated excellent linearity within the concentration ranges of 10–15,000 ng/mL and 20–10,000 ng/mL, respectively (Table 2). Good correlation coefficients, *r* values (amoxicillin, 0.9963 ± 0.00112; clavulanate, 0.9978 ± 0.000929), were exhibited during the validation assay (Table 2). The back-calculated concentrations of all the calibration curves were within ±15% of the nominal value: from −7.67 to 13.1% for amoxicillin and from −5.25 to 8.90% for clavulanate, respectively (Table 2).

#### 2.2.3. Carryover

After injecting the upper limit of quantitation (ULOQ) samples, no peaks corresponding to the two analytes or IS were observed at their respective retention times in the blank samples, confirming a lack of carryover.

#### 2.2.4. Accuracy and Precision

Table 3 presents the accuracy and precision for the LLOQ and three QC samples for each analyte. Accuracy was expressed as the % relative error (RE, %), while the precision was determined in terms of relative standard deviation (RSD, %). The intra-day % REs and % RSDs were 4.53 to 10.9% and ≤ 5.44% for amoxicillin; and 4.32 to 8.73% and ≤10.7% for clavulanate. The inter-day % REs and % RSDs were −1.26 to 10.2% and ≤7.08% for amoxicillin; and −4.41 to 2.13% and ≤8.74% for clavulanate. The intra- and inter-day accuracies and precisions all met the acceptance criteria [29].

#### 2.2.5. Matrix Effects

No significant matrix effects were observed for eight different human plasma lots, including lipemic or hemolyzed plasma (Table 4). The internal standard (IS) normalized matrix factor (MF) of amoxicillin and clavulanate were listed in Table 4 at three QC levels, respectively. This assay did not reveal matrix effects, ion suppression, or ion enhancement, considering that the relative standard deviation (RSD, %) values obtained were all smaller than 15% [30]. Therefore, the analysis was conducted reliably and with minimal matrix effects.

#### 2.2.6. Extraction Recovery

As presented in Table 5, the mean extraction recoveries were 86.7–87.4% (RSD ≤ 5.00%) for amoxicillin and 85.4–85.8% (RSD ≤ 4.85%) for clavulanate, demonstrating that all analytes were reproducibly and consistently recovered. In addition, the IS exhibited an extraction recovery of 87.9 ± 2.77%, with an RSD of 4.87%.

#### 2.2.7. Dilution Integrity

After a 10-fold dilution of QC sample (60/40 μg/mL for amoxicillin/clavulanate; six replicates) with pooled blank human plasma, the % RE and % RSD of diluted QCs were −11.3 to −3.33% and ≤2.83% for amoxicillin; and −8.50 to −1.50% and ≤2.70% for clavulanate. These findings satisfied the acceptance criteria [29], demonstrating that human plasma samples with concentrations up to 4-fold above the ULOQ can be reliably measured with a proper dilution.

#### 2.2.8. Stability

The stock solutions of amoxicillin and clavulanate were considered to be stable for 60 days at −80 °C, 4 h at room temperature (mean 25 ± 4 °C), and 24 h at 4 °C. Under all conditions, the peak response values showed RSD and RE values within 10% relative to those of freshly prepared stock solutions, with recoveries exceeding 94.1 ± 2.10%. While the amoxicillin stock solution was stable for up to 24 h at room temperature, the stock solution of clavulanate degraded after 6 h storage at room temperature, with recoveries of 88.9 ± 7.38%, 80.0 ± 3.38%, and 62.9 ± 4.51% after 6, 12, and 24 h, respectively.

Table 6 summarizes the detailed benchtop (for 4 h at ambient temperature), long-term (for 60 days at −80 °C), freeze–thaw (three cycles), and post-preparative (for 24 h in the 4 °C auto-sampler rack) stabilities of amoxicillin and clavulanate in human plasma. The accuracy (RE, %) and precision (RSD, %) of the stabilities were well within the acceptable limit [29], suggesting that there are no obvious problems in stability for sample processing and storage.

#### 2.2.9. Anticoagulant (Blood Collection-Tube Types) Effects

To assess the effects of different anticoagulants, plasma samples were collected using human lithium–heparin, sodium–heparin, and K_3_-EDTA anticoagulated tubes. The results of intra-assay accuracy and precision are summarized in Table 7. All the different tested anticoagulant QC samples were within the acceptable limits of accuracy (±15% RE) and precision (≤15% RSD) [29]: the %RE and %RSD were −7.63 to 5.72% and ≤8.40% for amoxicillin; and −7.37 to 6.48% and ≤8.90% for clavulanate. The three different anticoagulants tested had no impact on the concentration of the calibration curves and QC samples using human K_2_-EDTA plasma. This bioanalytical method demonstrated a consistent performance across plasma samples collected with lithium–heparin, sodium–heparin, and K_3_-EDTA, with no significant bias observed. These findings suggest that the tested blood collection-tube types are acceptable alternatives for pharmacokinetic sample collection.

#### 2.2.10. Clinical Application and Incurred Sample Reanalysis (ISR)

The validated bioanalytical methods were applied to the clinical pharmacokinetic study of Augmentin^®^ IR tablets in seventeen healthy Korean subjects successfully. The LLOQ of this assay (10/20 ng/mL) using a small plasma volume (100 μL) was sufficient for assessing pharmacokinetic behaviors of amoxicillin and clavulanate. No adverse events occurred during the entire study period. The mean plasma concentration—time profiles of amoxicillin and clavulanate after the administration of Augmentin^®^ IR tablet (250/125 mg) are depicted in Figure 3, and the relevant pharmacokinetic parameters are listed in Table 8.

Amoxicillin and clavulanate occurred at median T_max_ of 1.5 h, with mean C_max_ values of 5050 ± 1150 ng/mL and 2430 ± 1070 ng/mL, and mean AUC_t_ values of 12,200 ± 2240 ng∙h/mL and 5630 ± 2660 ng∙h/mL, respectively (Table 8). These pharmacokinetic parameters were comparable with earlier reported in the literature [8,11]. No statistically significant differences in all pharmacokinetic parameters of amoxicillin and clavulanate were observed between male (*n* = 12) and female (*n* = 5) subjects.

These results revealed significant inter-subject variability in the pharmacokinetic parameters of clavulanate among healthy Korean subjects. The inter-subject variability (CV, %) of clavulanate was 43.9% for C_max_ and 47.2% for AUC_t_, whereas that of amoxicillin was 22.7% and 18.2%, respectively (Table 8). Al-Sallami et al. [31] reported that inter-subject variance in pharmacokinetic parameters was considered “low” (CV% ≤ 10%), “moderate” (CV% ∼ 25%), or “high” (CV% > 40%). Based on this criterion, clavulanate belongs to be highly variable drug. High inter-subject variability in clavulanate absorption has been reported in clinical studies involving healthy volunteers [32,33].

This validated bioanalytical method was developed to support future drug–drug interaction (DDI) studies using the Augmentin^®^ IR tablet as a victim drug. In this pilot clinical study, this method was successfully applied to characterize the pharmacokinetics of amoxicillin and clavulanate following oral administration of the lowest-dose formulation (250/125 mg). Given that the amoxicillin calibration curve ranged from 10 to 15,000 ng/mL and was extendable to 60,000 ng/mL with dilution, and that the observed mean C_max_ was approximately 5050 ng/mL, the method is expected to be applicable to higher-dose formulations (e.g., 500/125 mg or 875/125 mg) that are frequently used in clinical practice.

Furthermore, since amoxicillin demonstrates non-linear pharmacokinetics due to saturation of oral absorption, leading to less than dose-proportional increases in exposure [8,34], the current method is likely to be suitable for quantifying a broad range of clinical doses within the validated calibration range. The high inter-individual variability observed in clavulanate exposure in this study provides additional insight for the design of future DDI studies. These findings may inform the estimation of appropriate sample sizes and the optimization of blood-sampling time points, particularly in trials evaluating potential interactions with therapeutic agents widely prescribed for infectious conditions, including upper respiratory tract infections.

The ISR results showed that all selected samples (total 51 plasma samples) from each subject met the acceptance criteria; % changes in concentrations between the initial and the reanalysis were all within ±20%, confirming the comparability and reliability of this assay during storage and processing of the clinical samples.

#### 2.2.11. Greenness Assessment

The Analytical GREEnness (AGREE) tool was employed to assess the greenness of this developed method using the AGREE calculator (version 0.5) (Figure 4) [35], as it provides a flexible and comprehensive evaluation [36] by integrating all 12 principles of GAC [37]. It generates both qualitative and quantitative outputs in a format that is user-friendly and readily interpretable [38]. This score supports its suitability as an eco-friendly analytical approach.

## 3. Materials and Methods

### 3.1. Chemicals and Materials

Amoxicillin (purity ≥ 90%); potassium clavulanate (purity ≥ 98%); formic acid (purity ≥ 95%); dimethyl sulfoxide (purity ≥ 99.7%); the internal standard, 4-hydroxytolbutamide (purity ≥ 98%; IS); and syringe filters (13 mm × 0.2 μm, PTFE membrane) were obtained from Sigma-Aldrich (St. Louis, MO, USA). LC–MS-grade acetonitrile, methanol, and dichloromethane were obtained from Burdick & Jackson Company (Morristown, NJ, USA). Water was purified in-house by a Milli-Q Ultrapure Water System (Millipore, Bedford, MA, USA). Other reagents used were of analytical grade or higher.

Blank human K_2_-EDTA plasma and lipemic plasma were purchased from BioChemed Services, Inc. (Winchester, VA, USA), and stored at −20 °C prior to use. Human whole blood was supplied from Clinical Pharmacology Center of Jeonbuk National University Hospital (Jeonju, Republic of Korea) and stored at 4 °C.

### 3.2. Instrumentation and Chromatographic Conditions

A Q-Trap 5500 mass spectrometer (Sciex, Framingham, MA, USA) in electrospray ionization (ESI) mode to generate negative species was coupled with an Agilent 1260 HPLC system (Agilent Technologies, Wilmington, DE, USA). The Poroshell 120 EC-C_18_ column (2.7 μm, 4.6 × 100 mm, Agilent Technologies, Wilmington, DE, USA) was used for sample separation. The mobile phase was composed of 0.1% aqueous formic acid (A) and acetonitrile (B), with a flow rate of 0.5 mL/min. The gradient elution was as follows: 0.0–0.2 min (90% A), 0.2–0.3 min (90–20% A), 0.3–1.0 min (20% A), 1.0–1.1 min (20–90% A), and 1.1–6.5 min (90% A). The autosampler was kept at 4 °C. The selected reaction monitoring (SRM) mode for quantitation was employed by setting at *m*/*z* 363.9→223.1 for amoxicillin, *m*/*z* 198.0→135.8 for clavulanate, and *m*/*z* 285.0→185.8 for the IS. The optimized ion-spray voltage and temperature were set at −4500 V and 600 °C, respectively. The following operating conditions were optimized based on the flow injection of amoxicillin, clavulanate, and the IS: declustering potential, collision energy, and entrance potential of −50 V, −13 V, and –10 V for amoxicillin; −20 V, −5 V, and −10 V for clavulanate; and −20 V, −18 V, and −10 V for the IS, respectively. Nitrogen gas was used as the nebulizer, curtain, and collision-activated dissociation gas, which were set to 20 psi, 10 psi, and medium, respectively. Analyst 1.5.2 software (Sciex, Framingham, MA, USA) was used to acquire and quantify data.

### 3.3. Preparation of the Calibration Standards and QC Samples

Stock solutions of amoxicillin and clavulanate were prepared at concentrations of 9 mg/mL and 6 mg/mL in dimethyl sulfoxide, respectively. For calibration curve construction, working solutions of each analyte were individually prepared by serial dilution of the respective stock solutions with distilled water, yielding concentrations ranging from 2–3000 μg/mL for amoxicillin and 4–2000 μg/mL for clavulanate. These concentrations correspond to 200-fold the target plasma levels. On the day of calibration, the separately stored working solutions were mixed in a 1:1 (*v*/*v*) ratio immediately prior to spiking. Calibration standards were prepared by spiking 1 μL of the mixed working solution into 99 μL of blank human plasma, resulting in final concentrations of 10/20, 50/100, 250/200, 500/500, 2500/1000, 7500/5000, and 15,000/10,000 ng/mL for amoxicillin and clavulanate, respectively.

Quality control (QC) samples were prepared in the same manner at four concentration levels: lower limit of quantification (LLOQ, 10/20 ng/mL), low QC (30/60 ng/mL), medium QC (6000/4000 ng/mL), and high QC (12,000/8000 ng/mL) for amoxicillin and clavulanate, respectively. All QC samples were independently prepared from separately diluted working solutions. Plasma calibration standards and QC samples were prepared freshly daily.

The stock solution of 4-hydroxytolbutamide (IS) was made by dissolving at 0.5 mg/mL in acetonitrile and further diluted finally to 20 ng/mL in acetonitrile for daily IS solution and stored at 4 °C. All the prepared stock was kept at −80 °C, and individual working solutions were stored at −20 °C.

### 3.4. Plasma Sample Preparation

Before the analysis, fresh or frozen plasma samples were thawed in an ice-water bath. Amoxicillin and clavulanate in the plasma samples were extracted by deproteinization with acetonitrile and then back-extracted using distilled water–dichloromethane (2:7, *v*/*v*).

In detail, an ice-cold 300 μL of IS solution (20 ng/mL 4-hydroxytolbutamide, dissolved in acetonitrile) was first added to 100 μL of human plasma and vortexed for 5 min. After centrifugation at 16,000× *g* for 10 min at 4 °C, 250 μL of the supernatant was transferred into a new 1.5 mL tube. Distilled water–dichloromethane (100 μL:350 μL, *v*/*v*) was then added to the supernatant, followed by vortexing for 10 min and centrifugation (16,000× *g*, 10 min, 4 °C). Finally, the aqueous upper layer (approximately 130 μL) was filtered directly into an autosampler vial through a syringe PTFE filter (13 mm × 0.2 μm). All procedures were carried out at 4 °C and completed within 60 min. A 5 μL aliquot was injected into the LC–MS/MS system for quantification.

### 3.5. Method Validation

The bioanalytical method validation for this assay was conducted following the regulatory guidance provide by the United States Food and Drug Administration (US FDA) and International Council for Harmonisation of Technical Requirements for Pharmaceuticals for Human Use (ICH) [29,30], covering parameters such as specificity, linearity, sensitivity, accuracy, precision, matrix effects, extraction recovery, dilution integrity, stability, and incurred sample reanalysis (ISR) of the analytes in human K_2_-EDTA plasma. In addition, the method’s applicability was assessed using plasma samples collected with various anticoagulants, including lithium–heparin, sodium–heparin, and K_3_-EDTA.

#### 3.5.1. Specificity and Sensitivity

To assess specificity, eight individual blank plasma samples, including one hemolyzed and one lipemic plasma, were analyzed in six replicates. This analysis confirmed the absence of endogenous interfering peaks at the respective retention times of amoxicillin, clavulanate, and IS at the LLOQ.

The LLOQ of the analytes was defined as the lowest concentration in the calibration ranges, with a signal-to-noise (S/N) ratio > 10; its accuracy should be within 80.0–120.0%, and its precision should not exceed 20.0% of the theoretical value when injected in six replicates.

#### 3.5.2. Linearity

The calibration curve was established by plotting the ratio of the peak area of each analyte to that of the IS (*y*) against the corresponding nominal concentration (*x*), ranging from 10 to 15,000 ng/mL for amoxicillin and 20 to 10,000 ng/mL for clavulanate. Linear regression with weighting factors (none, 1/*x*, or 1/*x*^2^) was applied to the data to determine the appropriate calibration model. The acceptance criteria for the seven non-zero back-calculated calibration curves’ concentrations were ± 15% from the nominal concentrations, except for LLOQ (±20%).

#### 3.5.3. Carryover

The injection of the highest concentration sample in the calibration range, ULOQ, followed by blank biological samples, was used to investigate carryover. For a blank sample, the % carryover should be less than 20% of the LLOQ and 5% of the IS peak area.

#### 3.5.4. Accuracy and Precision

The intra-day accuracy and precision were determined by analyzing six replicates of the LLOQ and three different QC levels (low, medium, and high) within the same day. The inter-day accuracy and precision were also assessed by measuring the LLOQ and the three QCs over five separate days, with two replicates per day. The calculated concentrations of the LLOQ and the three QCs were determined from the freshly prepared calibration standards and analyzed in the same run. Accuracy was expressed as the % relative error (RE, %), while the precision was measured in terms of % relative standard deviation (RSD, %). The accepted %RE values of three QCs were within ±15% (±20% for LLOQ), and the % RSD values should be within 15% (20% for LLOQ).

#### 3.5.5. Matrix Effect

The matrix effects of amoxicillin and clavulanate were evaluated by comparing the peak area in the presence of the matrix (post-extraction spiked plasma) with the peak area in the absence of the matrix (non-extracted neat solution) at three QC levels using eight different human plasmas, including lipemic or hemolyzed plasma (*n* = 3). A IS normalized matrix factor (MF) was calculated for each sample as follows: IS normalized MF = [(Analyte response in the post-extraction spiked matrix/analyte response in the neat solution standard)/(IS response in the post-extraction spiked matrix/IS response in the neat solution standard)]. The % RSD must be ≤ 15% [30].

#### 3.5.6. Extraction Recovery

The extraction recovery of amoxicillin and clavulanate was determined by comparing the peak area of the extracted sample with that of the post-extraction spiked plasma. The three QC levels of amoxicillin were 30, 6000, and 12,000 ng/mL, and the QC levels of clavulanate were 60, 4000, and 8000 ng/mL, respectively, tested in K_2_-EDTA pooled human plasma (*n* = 6). The extraction recovery of the IS at 20 ng/mL was tested using the same method.

#### 3.5.7. Dilution Integrity

Amoxicillin- and clavulanate-spiked human plasma samples at concentrations of 60 μg/mL and 40 μg/mL (4-fold the respective ULOQs) were prepared, directly diluted 10-fold with K_2_-EDTA pooled blank human plasma, and analyzed in six replicates. To meet the acceptance criteria, the accuracy (RE, %) of the dilution QC samples must fall within ±15% of the nominal concentration, while the precision (RSD, %) should not exceed 15%.

#### 3.5.8. Stability

The stability of stored stock solutions of amoxicillin and clavulanate was evaluated at room temperature (4 h, 6 h, 12 h, and 24 h), 4 °C (24 h), and −80 °C (60 days), respectively. The peak area of stored samples was compared with those of fresh stock solutions; if the deviations were within ±10%, the stock solutions were considered stable [28].

The stabilities evaluation in human plasma for amoxicillin and clavulanate were performed in six replicates at three QC levels. The QCs were examined when kept at room temperature (4 h), −80 °C (60 days), 4 °C (24 h, processed QC sample), and up to three freeze–thaw cycles. The stability evaluation was performed by comparing the analyzed QC samples with freshly prepared calibration standards and QC samples. The samples were considered stable if the assay results were within the acceptable criteria of accuracy (±15% RE) and precision (≤15% RSD).

#### 3.5.9. Anticoagulant (Blood Collection-Tube Types) Effects

According to the regulatory guidelines, modifications to a validated bioanalytical method—such as the use of plasma as the matrix and a change in the anticoagulant from K_2_-EDTA to lithium heparin, sodium heparin, or K_3_-EDTA—require partial validation, which includes the assessment of intra-day accuracy and precision [29,30]. In this study, accuracy and precision were evaluated at three QC levels (30/60 ng/mL, 6000/4000 ng/mL, and 12,000/8000 ng/mL for amoxicillin and clavulanate), with six replicates prepared for each concentration. The concentrations of amoxicillin and clavulanate for the QC samples mentioned above were determined from the freshly prepared calibration standards, along with fresh QC (K_2_-EDTA plasma) samples, using K_2_-EDTA pooled human plasma. The % RE and % RSD should not exceed 15% [29,30].

#### 3.5.10. Clinical Application and ISR

This assay was used to evaluate pharmacokinetic samples from seventeen healthy Korean subjects (twelve males and five females) orally administered Augmentin^®^ IR tablet. This clinical trial was approved by the Institutional Review Board of Jeonbuk National University Hospital (Jeonju, Republic of Korea; IRB No. 2022-04-041-008). Prior to inclusion in the study, written informed consent was obtained from all participants. This clinical trial was designed to be randomized and open-labeled, with a single oral dosing, and conducted under a fasting condition. Seventeen healthy subjects were recruited according to the following inclusion criteria: 19–50 years old, with a body mass index (BMI) of 17.5–30.5 kg/m^2^. Participants with a history of hypersensitivity to amoxicillin/clavulanate and β-lactam antibiotics (e.g., penicillins and cephalosporins) were excluded prior to the start of the clinical trial. Study participants were identified as healthy based on laboratory testing and physical examinations. The participants received one Augmentin^®^ IR tablet (250/125 mg) with 240 mL of drinking water in a fasting state. Blood samples (~2 mL) were collected into chilled K_2_-EDTA tubes at pre-dose (0 h) and 0.25, 0.5, 0.75, 1, 1.5, 2, 2.5, 3, 4, 6, 8, and 10 h after the dosing. Then, the chilled blood samples were centrifuged immediately at 3000× *g* for 10 min at 4 °C to obtain plasma, and promptly stored at −80 °C prior to transport to the LC–MS/MS analytical laboratory. The pharmacokinetic parameters of amoxicillin and clavulanate were estimated via non-compartmental analysis using Phoenix^®^ WinNonlin^®^ (version 6.0; Certara USA, Princeton, NJ, USA). The maximum plasma concentration (C_max_) and time to reach C_max_ (T_max_) values were read directly from experimental observations.

To verify the reproducibility of the assay, ISR was conducted. Three samples from each volunteer were selected: one collected close to the C_max_, and the others during the elimination phase (approximately 4−8 h) for amoxicillin and clavulanate. As per the guidance, the % change in concentrations between the initial and the reanalysis value should be less than ±20% of their means for at least two-thirds of the repeats [29].

#### 3.5.11. Greenness Assessment

Various tools have been developed to assess the environmental friendliness of analytical methods, including AGREE, AGREEprep, the Green Analytical Procedure Index (GAPI), Complex GAPI, the National Environmental Methods Index (NEMI), and the Analytical Eco-scale. AGREE was considered the most suitable tool, as it systematically incorporates all 12 principles of GAC into a unified score and facilitates clear interpretation of the environmental sustainability of the method [37]. The environmental friendliness of our established LC–MS/MS method was assessed quantitatively using the AGREE calculator (version 0.5).

## 4. Conclusions

In this study, a sensitive LC–MS/MS method with simple sample preparation was developed and fully validated for simultaneously quantifying amoxicillin and clavulanate in human K_2_-EDTA plasma. In addition, we partially validated this assay using human lithium–heparin, sodium–heparin, and K_3_-EDTA plasma to evaluate its compatibility with various anticoagulants, supporting its applicability in multicenter and large-scale clinical studies. Compared to previously reported methods, the present assay offers improved sensitivity, a wide calibration curve range, and eco-friendliness, as confirmed by the AGREE tool. The method was successfully applied to characterize the pharmacokinetics of amoxicillin and clavulanate following oral administration of the lowest-dose Augmentin^®^ IR tablet (250/125 mg) in healthy Korean subjects. The high inter-individual variabilities in pharmacokinetic parameters of clavulanate were significantly greater than those of amoxicillin, and this finding may inform the clinical design of future drug–drug interaction.

## Figures and Tables

**Figure 1 pharmaceuticals-18-00998-f001:**
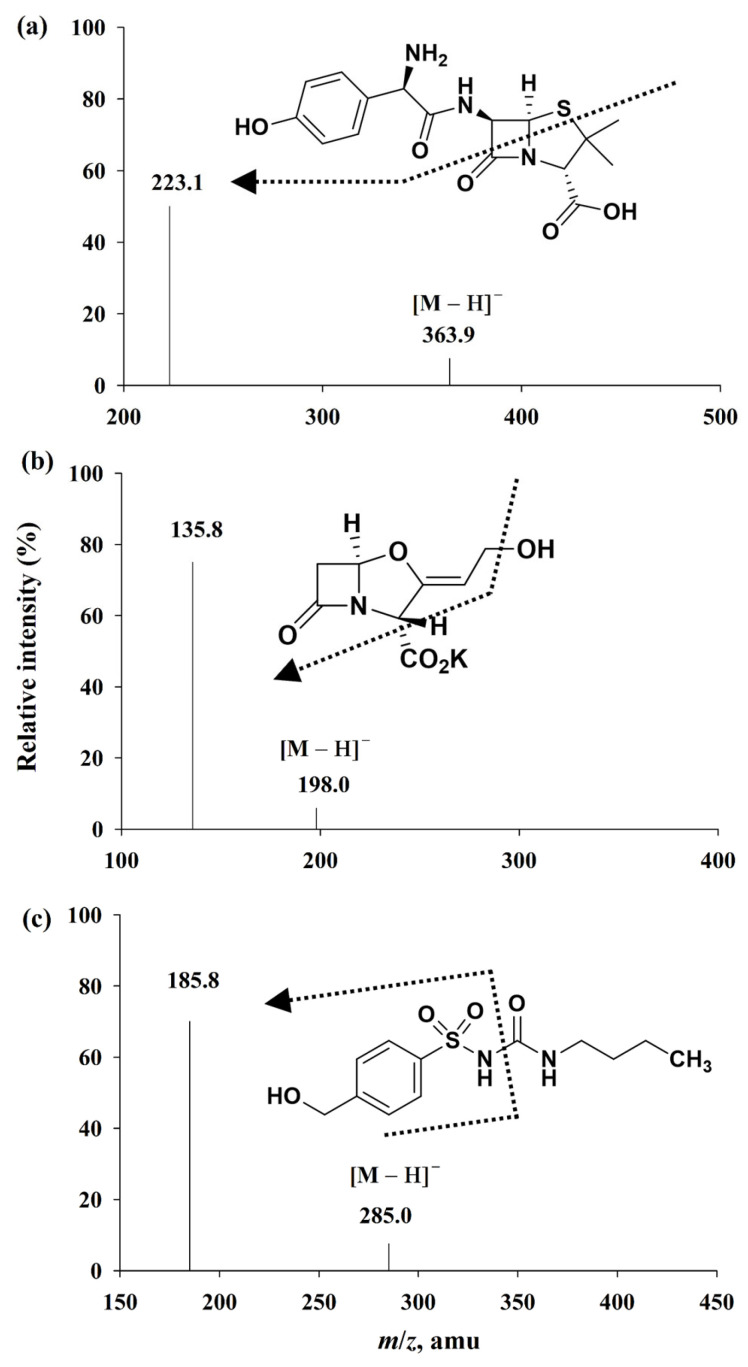
Chemical structures and product ion mass spectra of (**a**) amoxicillin, (**b**) clavulanate, and (**c**) 4-hydorxytolbutamide (the IS). The arrows indicate the selected product ion values (*m*/*z*) for each analyte.

**Figure 2 pharmaceuticals-18-00998-f002:**
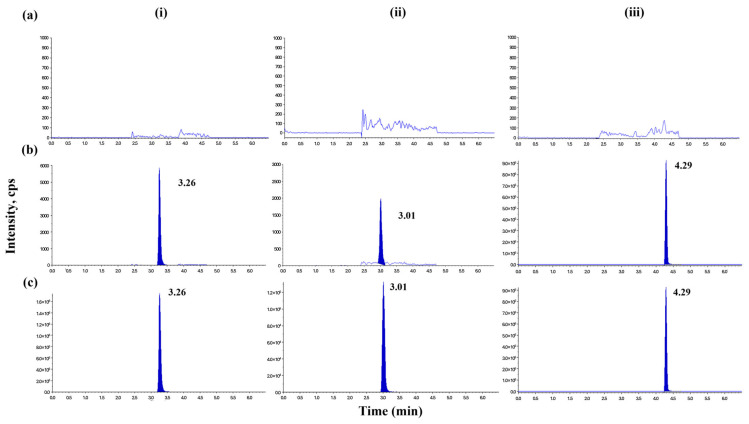
Typical LC–MS/MS chromatograms of (**i**) amoxicillin, (**ii**) clavulanate, and (**iii**) the IS: (**a**) blank plasma, (**b**) blank plasma spiked with both analytes at LLOQs (10/20 ng/mL for amoxicillin/clavulanate) and IS, and (**c**) a plasma sample obtained from a volunteer 2 h after a single oral dose of Augmentin^®^ (250/125 mg) IR tablet.

**Figure 3 pharmaceuticals-18-00998-f003:**
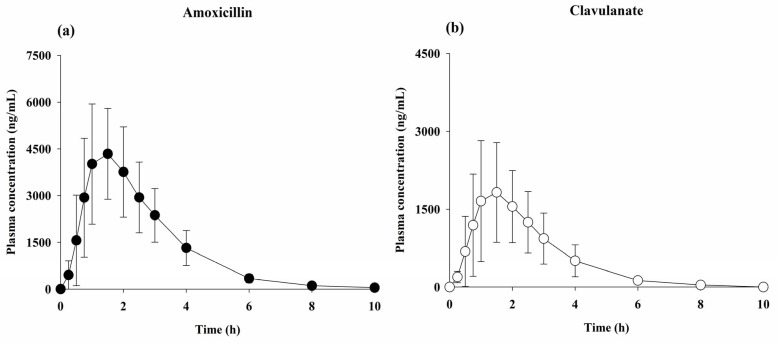
Mean plasma concentration—time profiles of (**a**) amoxicillin (●) and (**b**) clavulanate (○) after a single oral administration of Augmentin^®^ IR tablet (250/125 mg) in healthy Korean subjects (*n* = 17). Vertical bars represent standard deviation.

**Figure 4 pharmaceuticals-18-00998-f004:**
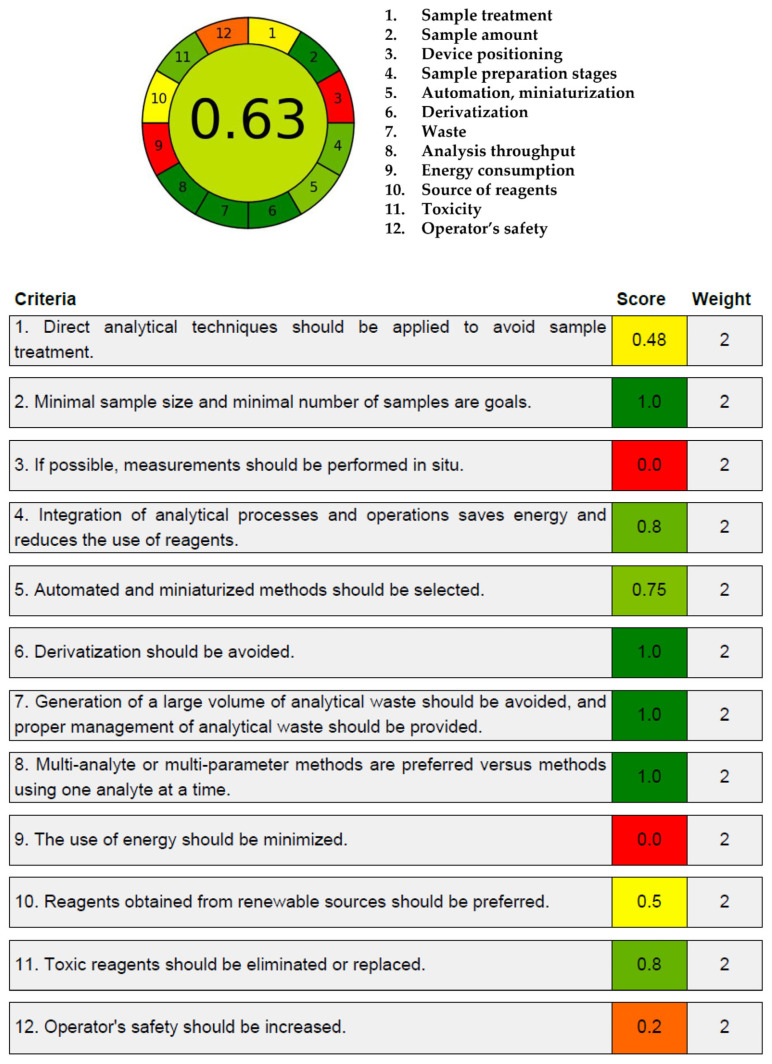
The AGREE pictogram for this bioanalytical method greenness representation.

**Table 1 pharmaceuticals-18-00998-t001:** Summary of the simultaneous quantification methods for amoxicillin and clavulanate in biological samples.

Analytes	Instrument	Matrices	Sample Volume (µL)	Internal Standard	Sample Preparation	Injection Volume (µL)	Calibration Range (ng/mL)	Run Time (min)	Full Validation	Reference
Amoxicillin	LC–MS/MS	Human plasma	100	4-hydroxytolbutamide	PP followed by LLE	5	10–15,000	6.5	Yes	PM
Clavulanate							20–10,000			
Amoxicillin	HPLC	Human plasma	100	none	PP	20	625–20,000	10	n/a	[9]
Clavulanate							312.5–10,000			
Amoxicillin	HPLC	Human plasma	500	Allopurinol	PP followed by LLE	50	200–12,000	10	n/a	[10]
Clavulanate							147–4908			
Amoxicillin	LC–MS	Human plasma	200	Terbutaline	PP followed by LLE	2	125–8000	3	n/a	[11]
Clavulanate							62.5–4000			
Amoxicillin	LC–MS	Human plasma	200	Carbamazepine	PP	4	10–40,000	5	n/a	[12]
Clavulanate							10–10,000			
Amoxicillin	LC–MS/MS	Human plasma	450	Ampicillin	PP with LLE	15	190–22,222	5	n/a	[13]
Clavulanate							147–4908			
Amoxicillin	LC–MS/MS	Human plasma	200	Diclofenac	PP with LLE	none	500–40,000	none	n/a	[14]
Clavulanate							100–6000			
Amoxicillin	LC–MS/MS	Human plasma	250	Hydrochlorothiazide	SPE and evaporation	10	103–6822	2	n/a	[15]
Clavulanate							46–3026			
Amoxicillin	LC–MS/MS	Human plasma	200	Amoxicillin-d_4_	SPE and evaporation	10	50–31,500	1.5	n/a	[16]
Clavulanate				Ampicillin			25–6000			
Amoxicillin	LC–MS/MS	Human plasma	200	Ampicillin	SPE and evaporation	5	50–10,000	2.4	Yes	[17]
Clavulanate							25–5000			
Amoxicillin		Human urine		Amoxicillin-d_4_			50–10,000			
Clavulanate				Sulbactam			50–10,000			
Amoxicillin	LC–MS/MS	Human plasma	100	Amoxicillin-d_4_	PP followed by LLE	50	3125–125,000	10.25	n/a	[18]
Clavulanate							1000–40,000			
Amoxicillin		Human tissue	5 (mg)	Amoxicillin-d_4_	PP	100	200–25,000			
Clavulanate				Flucloxacillin-^13^C_4_^15^N			200–25,000			
Amoxicillin	LC–MS/MS	Human plasma	50	Amoxicillin-d_4_	PP followed by LLE and evaporation	5	40–5000	4	Yes	[19]
Clavulanate							30–2500			
Amoxicillin	LC–MS/MS	Dog plasma	200	none	PP	none	0.5–500	12	n/a	[20]
Clavulanate							0.5–500			
Amoxicillin	LC–MS/MS	Human plasma	100	Clenbuterol hydrochloride	PP followed by LLE	10	5–16,000	5.5	n/a	[21]
Clavulanate							50–2000			
Amoxicillin	LC–MS/MS	Human plasma	50	Amoxicillin-d_4_	LLE and evaporation	5	20–5000	2	n/a	[22]
Clavulanate							10–2500			

HPLC, high-performance liquid chromatography; LC–MS, liquid chromatography–mass spectrometry; LC–MS/MS, liquid chromatography–tandem mass spectrometry; LLE, liquid–liquid extraction; n/a, not applicable; none, no data; PM, present method; PP, protein precipitation; SPE, solid-phase extraction.

**Table 2 pharmaceuticals-18-00998-t002:** Inter-day back-calculated concentrations of amoxicillin and clavulanate from their respective calibration standards in human plasma (*n* = 6).

Amoxicillin				Clavulanate			
Calibration Concentrations (ng/mL)	Back-Calculated (Mean ± SD; ng/mL)	Precision (RSD, %)	Accuracy (RE, %)	Calibration Concentrations (ng/mL)	Back-Calculated (Mean ± SD; ng/mL)	Precision (RSD, %)	Accuracy (RE, %)
10	9.83 ± 0.113	1.14	1.68	20	19.7 ± 0.264	1.34	1.42
50	53.2 ± 2.65	4.98	−6.37	100	104 ± 5.57	5.38	−3.57
250	269 ± 11.6	4.30	−7.67	200	211 ± 8.60	4.08	−5.25
500	525 ± 14.0	2.67	−4.93	500	511 ± 16.6	3.25	−2.20
2500	2550 ± 120	4.72	−1.93	1000	1040 ± 46.5	4.49	−3.60
7500	7040 ± 210	2.98	6.13	5000	4780 ± 107	2.23	4.33
15,000	13,000 ± 137	1.05	13.1	10,000	9110 ± 201	2.21	8.90
Slope	0.000715 ± 0.0000513	–	–	Slope	0.000125 ± 0.0000108	–	–
Intercept	0.000838 ± 0.000493	–	–	Intercept	0.000760 ± 0.000310	–	–
*r*	0.9963 ± 0.00112	–	–	*r*	0.9978 ± 0.000929	–	–

SD, standard deviation; RSD, relative standard deviation; RE, relative error; *r*, correlation coefficient.

**Table 3 pharmaceuticals-18-00998-t003:** Intra- and inter-day accuracy and precision of QCs for amoxicillin and clavulanate in K_2_-EDTA pooled human plasma.

Analytes	Spiked Concentration (ng/mL)	Intra-Day (*n* = 6)			Inter-Day (*n* = 10)		
		Mean ± SD (ng/mL)	Accuracy (RE, %)	Precision (RSD, %)	Mean ± SD (ng/mL)	Accuracy (RE, %)	Precision (RSD, %)
Amoxicillin	10	8.91 ± 0.476	10.9	5.34	8.98 ± 0.535	10.2	5.95
	30	27.1 ± 1.47	9.88	5.44	28.3 ± 2.00	5.69	7.08
	6000	5730 ± 181	4.53	3.15	6070 ± 318	−1.26	5.24
	12,000	10,700 ± 343	10.6	3.20	11,100 ± 263	7.91	2.38
Clavulanate	20	19.1 ± 2.04	4.75	10.7	19.6 ± 1.71	2.13	8.74
	60	56.5 ± 2.68	5.82	4.74	62.3 ± 4.25	−3.67	6.83
	4000	3830 ± 140	4.32	3.64	4180 ± 266	−4.41	6.37
	8000	7300 ± 291	8.73	3.99	7840 ± 256	1.98	3.27

SD, standard deviation; RE, relative error; RSD, relative standard deviation.

**Table 4 pharmaceuticals-18-00998-t004:** Matrix effect for amoxicillin and clavulanate in eight different human plasma (*n* = 3).

Amoxicillin									
Plasma Lot	30 ng/mL			6000 ng/mL			12,000 ng/mL		
	Accuracy (RE, %)	RSD, %	IS Normalized MF	Accuracy (RE, %)	RSD, %	IS Normalized MF	Accuracy (RE, %)	RSD, %	IS Normalized MF
Lot-1	98.9	5.94	1.00	105	5.39	0.959	92.9	0.990	0.932
Lot-2	94.1	10.5	0.927	96.8	3.65	0.922	91.4	0.774	0.876
Lot-3	92.9	1.83	1.00	99.7	4.75	0.983	95.0	0.595	0.911
Lot-4	89.0	5.08	1.03	101	2.66	1.01	89.1	0.159	0.930
Lot-5	96.8	12.1	1.07	104	9.88	0.993	92.1	0.921	0.958
Lot-6	90.8	2.26	1.04	96.2	2.50	0.994	92.1	2.30	0.959
Lot-7	87.9	3.38	1.08	94.0	0.677	1.09	89.9	1.81	1.07
Lot-8	91.8	10.5	0.984	96.3	5.51	1.09	86.2	0.820	1.10
**Clavulanate**									
**Plasma Lot**	**60 ng/mL**			**4000 ng/mL**			**8000 ng/mL**		
	**Accuracy** **(RE, %)**	**RSD, %**	**IS Normalized MF**	**Accuracy** **(RE, %)**	**RSD, %**	**IS Normalized MF**	**Accuracy** **(RE, %)**	**RSD, %**	**IS Normalized MF**
Lot-1	108	6.55	1.01	112	6.98	1.02	99.4	2.28	0.990
Lot-2	103	8.53	0.929	98.6	1.65	0.934	96.8	0.365	0.903
Lot-3	107	1.99	0.964	103	5.42	1.00	103	2.07	0.965
Lot-4	97.7	1.23	0.981	105	1.35	1.02	96.9	1.90	0.962
Lot-5	103	13.1	1.02	104	11.0	1.04	94.6	1.12	1.00
Lot-6	92.4	2.60	1.05	98.2	2.67	1.04	94.0	5.57	1.02
Lot-7	96.1	7.21	1.09	93.1	0.608	1.12	91.3	1.16	1.12
Lot-8	94.1	5.79	1.02	95.8	5.46	1.13	88.6	2.95	1.12

CV, coefficient of variation; MF, matrix factor.

**Table 5 pharmaceuticals-18-00998-t005:** Extraction recovery for amoxicillin, clavulanate, and 4-hydroxytolbutamide in K_2_-EDTA pooled human plasma (*n* = 6).

Analytes	Spiked Concentration (ng/mL)	Extraction Recovery (%)	
		Mean ± SD	RSD
Amoxicillin	30	87.4 ± 4.36	5.00
	6000	86.7 ± 3.03	3.50
	12,000	87.4 ± 3.68	4.21
Clavulanate	60	85.7 ± 4.15	4.85
	4000	85.4 ± 2.91	2.91
	8000	85.8 ± 3.53	3.53
4-Hydroxytolbutamide	20	87.9 ± 2.77	4.87

SD, standard deviation.

**Table 6 pharmaceuticals-18-00998-t006:** Stability of amoxicillin and clavulanate in K_2_-EDTA pooled human plasma under the various conditions (*n* = 6).

Analytes	Spiked Concentration (ng/mL)	Benchtop (4 h, Ambient)		Long-Term (60 Days, −80 °C)		Freeze–Thaw (3 Cycles)		Post-Preparative (24 h at 4 °C, Autosampler)	
		RE (%)	RSD (%)	RE (%)	RSD (%)	RE (%)	RSD (%)	RE (%)	RSD (%)
Amoxicillin	30	5.28	4.99	3.56	3.78	12.8	2.25	−7.46	4.86
	6000	5.58	2.22	7.56	10.2	11.2	1.11	3.33	3.35
	12,000	6.99	2.17	13.0	6.49	13.1	3.37	12.0	2.49
Clavulanate	60	12.5	2.48	2.41	8.78	13.8	4.19	−3.10	6.32
	4000	14.0	3.61	7.64	8.30	12.3	2.15	6.48	2.13
	8000	14.0	1.69	12.3	7.34	14.3	3.78	11.8	2.87

RE, relative error; RSD, relative standard deviation.

**Table 7 pharmaceuticals-18-00998-t007:** Intra-day accuracy and precision of three QCs for amoxicillin and clavulanate in lithium– heparin, sodium–heparin, and K_3_-EDTA pooled human plasma (*n* = 6).

Anticoagulants	Analytes	Spiked Concentration (ng/mL)	Intra-Day (*n* = 6)		
			Mean ± SD (ng/mL)	Accuracy (RE, %)	Precision (RSD, %)
Lithium–heparin	Amoxicillin	30	30.9 ± 0.999	3.14	3.23
		6000	5900 ± 266	−1.62	4.51
		12,000	11,100 ± 554	−7.63	5.00
	Clavulanate	60	58.2 ± 2.44	−2.90	4.19
		4000	3940 ± 252	−1.53	6.41
		8000	7830 ± 487	−2.20	6.22
Sodium–heparin	Amoxicillin	30	29.8 ± 1.30	−0.806	4.35
		6000	5990 ± 151	−0.133	2.52
		12,000	11,860 ± 996	−1.15	8.40
	Clavulanate	60	63.6 ± 2.75	6.48	4.33
		4000	3950 ± 73.4	−1.16	1.86
		8000	7990 ± 352	−0.217	4.40
K_3_-EDTA	Amoxicillin	30	31.8 ± 1.26	5.72	3.97
		6000	6015 ± 365	0.417	6.08
		12,000	11,700 ± 596	−3.00	5.11
	Clavulanate	60	59.6 ± 5.30	−1.41	8.90
		4000	3800 ± 148	−7.37	3.88
		8000	7760 ± 398	−3.10	5.13

RE, relative error; RSD, relative standard deviation.

**Table 8 pharmaceuticals-18-00998-t008:** Pharmacokinetic parameters (mean ± SD) of amoxicillin and clavulanate after oral administration of Augmentin^®^ IR tablet (250/125 mg) in healthy Korean subjects (*n* = 17).

Parameters	Amoxicillin	Clavulanate
	Mean ± SD	Mean ± SD
AUC_t_ (ng∙h/mL)	12,200 ± 2240	5630 ± 2660
AUC_inf_ (ng∙h/mL)	12,300 ± 2250	5680 ± 2650
C_max_ (ng/mL)	5050 ± 1150	2430 ± 1070
T_max_ (h)	1.5 [0.5–2.0]	1.5 [1.0–2.5]
Terminal t_1/2_ (h)	1.26 ± 0.188	0.972 ± 0.105

AUC_t_, area under the plasma concentration–time curve from time zero to time last sampling time; AUC_inf_, area under the plasma concentration–time curve from time zero to infinity; C_max_, maximum plasma concentration; T_max_, time to reach C_max_ (median [ranges]); Terminal t_1/2_, terminal half-life; SD, standard deviation.

## Data Availability

The data presented in this study are available in the article. The data will be made available upon request.

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
