# Peer review of "Development and Validation of Bioanalytical LC–MS/MS Method for Pharmacokinetic Assessment of Amoxicillin and Clavulanate in Human Plasma"

_pharmaceuticals, 2025, doi:10.3390/ph18070998_

Round 1

Reviewer 1 Report

Comments and Suggestions for Authors

The authors present a validated LC-MS/MS method for simultaneous quantification of amoxicillin and clavulanate in human plasma, emphasizing simplicity, sensitivity, and applicability to pharmacokinetic (PK) studies. However, many revisions are required particularly concerning novelty and merits.

  1. Regarding choice of internal standard. Why the authors choose hydroxytolbutamide  and not ampicilin as in the following article https://link.springer.com/article/10.1186/s12917-025-04649-4?
  2. The authors should illustrate the reasons for choosing such IS.
  3. In the abstract, the authors should discuss in details limitations of the old methods and merits for the new one .
  4. In the abstract,  add brief PK metrics (e.g., Cmax, t1/2) to demonstrate method utility in generating clinically meaningful data.
  5. In a table form,Compare the method directly with 2 or 3 recent publications to highlight specific advancements. Emphasize any unresolved challenges in prior work that this method resolves.
  6. No mention of measures to address clavulanate instability (e.g., acidification, storage conditions), which is critical for method credibility.
  7. In line 130, mention the creteria for selection of this IS.
  8. In line 398, Korean subjects (twelve males and five females) ; is there effect for gender on results ? discuss in details
  9. Add future plans and study limitation
  10. For the conclusion, add merits for the novel method over old ones and any new results in pharmacokinetic data for the Korean people.

Reviewer 2 Report

Comments and Suggestions for Authors

Huge effort was exerted, nevertheless lacking novelty and some main points.

1- The work unfortunately lacks novelty where several papers published similar work and authors did not show convincing comparison to them, additionally, they missed some of previously published work for this mixture and did not compare to it like:

  1. Simultaneous Determination of Amoxicillin and Clavulanic Acid by LC-MS/MS in Human Plasma

Ji Shunli et al., Published in 2019 in Journal of China Pharmaceutical University

This research focused on developing a sensitive LC-MS/MS method for the simultaneous determination of amoxicillin and clavulanic acid in human plasma. The method was successfully applied to a bioequivalence study.

https://pesquisa.bvsalud.org/portal/resource/pt/wpr-807918

  1. A 2D HPLC-MS/MS Method for Several Antibiotics in Blood Plasma and Tissue Samples

Rehm et al, Published in 2020 in Analytical and Bioanalytical Chemistry

This study presented a two-dimensional high-performance liquid chromatography–tandem mass spectrometry (2D HPLC-MS/MS) method for quantifying various antibiotics, including amoxicillin and clavulanic acid, in human plasma and tissue samples. ​

https://pubmed.ncbi.nlm.nih.gov/31900530/

2- the mobile phase used is typically not green, where formic acid is used, and it is common now to present ecofriendly analyses and care for this. Green analyses are the trending topics of analysis now. Authors should present greenness assessments of their new method e.g. using AGREE metric.

3- application of the method is limited to just one product: (This method was sufficiently able to evaluate pharmacokinetics of Augmentin® immediate-release tablet (250/125 mg) in humans.), while there are several products and several formulas with different concentrations for this mixture.

4- authors mention: linear concentration ranges were 10–15,000 ng/mL for amoxicillin (r ≥ 0.9945) and 20–27 10,000 ng/mL for clavulanate, despite showing their method as very sensitive, they mixed up meanings of linearity with calibration range, where ICH guidelines mention them as two separate terms and the practical range must shows high precision for all its values, while 10 and 20 ng for sure will lack this precision on replication.

5- no clear method validation protocol like ICH guidelines or USP and not all validation parameters were covered in this study.

Reviewer 3 Report

Comments and Suggestions for Authors

Authors have developed and validated bionalytical method to simultaneously quantify antibiotic drugs. It is a good internal memo for peoples working on these compounds for clinical applications. However I have several doubts and suggestions regarding study:

  1. Authors have not mentioned injection volume during their LC-MS analysis. Also the effect of it on sensitivity.
  2. I am concerned about extraction procedure as they have not mentioned the optimization of extraction solvents. I am suggesting use of 0.1% Formic acid in acetonitrile for a simple protein precipitation. Let the reader know difficulties in simple extraction procedures.
  3. Authors are suggested to provide all pharmacokinetic parameters after NCA analysis including Vd, Clearance, MRT.
  4. Suggesting to provide any specific reason for high variation in plasma concentrations among subjects. Explanation from the literature will work.
  5.  In material method section 3.3 please explain how dilutions are prepared as if the stocks are prepared by serial dilutions and concentration are not matching when they diluted eg. 15000/10000 >2x> 7500/5000 >5x> 2500/1000 here 5x dilution doesn't matches if mixture is serially diluted.
  6.  Reconsider doing matrix effect as mentioned in validation draft ICH M10/USFDA guidance.

Round 2

Reviewer 1 Report

Comments and Suggestions for Authors

The author did all required changes. the paper could be published in the current form. 

Reviewer 2 Report

Comments and Suggestions for Authors

thanks for the efforts done to improve the work, it seems better now

Reviewer 3 Report

Comments and Suggestions for Authors

The authors have addressed all the comments and suggestions from review in a thorough and satisfactory manner. The revised manuscript has improved in clarity, structure, and scientific rigor.